# Controlled Decoration of [60]Fullerene with Polymannan Analogues and Amino Acid Derivatives through Malondiamide-Based Linkers

**DOI:** 10.3390/molecules27092776

**Published:** 2022-04-27

**Authors:** Lisa Tanzi, Davide Rubes, Teodora Bavaro, Matthieu Sollogoub, Massimo Serra, Yongmin Zhang, Marco Terreni

**Affiliations:** 1Department of Drug Sciences, University of Pavia, viale Taramelli 12, I-27100 Pavia, Italy; lisa.tanzi01@universitadipavia.it (L.T.); davide.rubes01@universitadipavia.it (D.R.); teodora.bavaro@unipv.it (T.B.); marco.terreni@unipv.it (M.T.); 2Institut Parisien de Chimie Moléculaire, Sorbonne Université, CNRS, Place Jussieu 4, 75005 Paris, France; matthieu.sollogoub@sorbonne-universite.fr

**Keywords:** bis-functionalized fullerenes, amino acid conjugates, glycofullerenes, mannose-based oligosaccharides, cyclopropanation

## Abstract

In the last few years, nanomaterials based on fullerene have begun to be considered promising tools in the development of efficient adjuvant/delivery systems for vaccination, thanks to their several advantages such as biocompatibility, size, and easy preparation and modification. In this work we reported the chemoenzymatic synthesis of natural polymannan analogues (di- and tri-mannan oligosaccharides characterized by α1,6man and/or α1,2man motifs) endowed with an anomeric propargyl group. These sugar derivatives were submitted to 1,3 Huisgen dipolar cycloaddition with a malondiamide-based chain equipped with two azido terminal groups. The obtained sugar-modified malondiamide derivatives were used to functionalize the surface of Buckminster fullerene (C_60_) in a highly controlled fashion, and yields (11–41%) higher than those so far reported by employing analogue linkers. The same strategy has been exploited to obtain C_60_ endowed with natural and unnatural amino acid derivatives. Finally, the first double functionalization of fullerene with both sugar- and amino acid-modified malondiamide chains was successfully performed, paving the way to the possible derivatization of fullerenes with immunogenic sugars and more complex antigenic peptides.

## 1. Introduction

Buckminster fullerene (C_60_), the smallest stable fullerene (diameter 10.34 Å), is characterized by high symmetry and two different types of carbon–carbon bonds. The 5,6 bond is a single bond of 1.45 Å length between 5-member and 6-member rings, whilst the 6,6 bond of 1.38 Å between two 6-member rings has a double bond character and is usually involved in fullerene reactivity.

In the last years, fullerenes have attracted attention for their possible therapeutic applications in humans. Pure C_60_ is soluble only in organic solvent and can present potentially high toxicity, but these limitations could be overcome by submitting fullerene to chemical modifications with polar biomolecules. There are three main synthetic approaches to gain these functionalized derivatives, i.e., by exploiting cyclopropanation, hydroxylation and cycloaddition reactions [1].

Depending on the conjugated molecule, fullerene derivatives acquire different therapeutic properties [2,3]. Conjugation of fullerene with carbohydrates for biological application has been largely investigated [4]. The most explored reaction to decorate C_60_ with sugar derivatives is cyclopropanation. Exploiting a Bingel cyclopropanation, up to six malonate ester chains carrying azido or propargyl terminal groups were one-pot linked to fullerene [5]. However, other conjugation methods were deeply investigated [6,7,8,9,10].

The effect of glycosylation on biological activity of fullerene is defined “indirect” if the role of the sugars is to improve fullerene solubility in water or if they are involved in targeted drug delivery. Glycofullerene derivatives have been investigated in photodynamic therapy [11,12,13], in HIV-1 protease inhibition [14,15,16] or against neurodegenerative diseases [17,18].

On the contrary, the role of the sugars can be considered “direct” if they have a defined therapeutic activity. In this case, the “multivalency” is the essential property that usually ensures a good biological activity of sugar-modified fullerene, e.g., these derivatives have been investigated as potential ligands for receptors and enzymes used by viruses and bacteria (e.g., *E. coli*, Ebola or Dengue viruses) during the infection process [19,20,21,22].

Similarly, the conjugation of amino acids or peptides with fullerene [1,23,24] provides structural diversity and specific recognition properties [25]. Amino acid derivatives of fullerenes have been mainly studied as anti-infective agents against viruses such as cytomegalovirus and HIV [26,27,28]. Fullerene conjugated with amino acids or peptides were also tested in the treatment of different pathologies that still require efficient therapeutic approaches, such as Alzheimer [29], lupus [30] and in cancer diagnosis [31,32].

In the last few years, thanks to their intrinsic properties, such as size, biocompatibility, and the possibility to undergo chemical modification, nanomaterials have attracted increasing attention as promising tools for the development of new generation vaccines [33,34]. Some studies reported the efficiency of C_60_ derivatives as vaccine adjuvants [35,36], suggesting the idea that fullerene could be exploited as a carrier for covalent conjugation of immunogenic or antigenic sugars and/or peptides.

The high expression of Mannose Receptors (MR) on Antigen Presenting Cells (APCs) indicates that MR play a key role in antigen recognition. Indeed, mannosylated peptides and proteins stimulate MHC class II specific T cells with 200 to 10,000-fold higher efficiency compared to peptides or proteins that are not mannosylated [37]. Fairbanks et al. [38] reported a stronger immune response obtained after the mannosylation of a peptide-epitope of a cytomegalovirus antigen compared to the non-glycosylated peptide.

Since our research activity is focused on the design of new glycovaccines, we considered the use of fullerene as a potential carrier for both immunogenic/antigenic sugars and peptides for the development of new generation nano-vaccines.

In this work we reported the decoration of C_60_ core by using malondiamide-based linkers endowed with natural polymannan analogues and natural/unnatural amino acid derivatives. The derivatization strategy is based on the conversion of fullerene into methanofullerenes through a Bingel cyclopropanation reaction, which usually takes place between fullerene and a halide-activated malonic acid derivative in the presence of a base [39].

Malondiamide-based linkers were chosen due to their higher chemical stability with respect to the corresponding ester derivatives. As a title of example, amide bond is stable under the basic reaction conditions required for the removal of acetyl protecting groups from sugar derivatives obtained through chemoenzymatic approaches, thus allowing the obtainment of fully deprotected sugar bioconjugates. Nevertheless, C_60_ functionalization with diamide-based linkers is a less used approach due to their low reactivity if compared with ester-based analogues. Among diamide-based linkers, those characterized by an aromatic amide moiety turned out to be more reactive than aliphatic ones. For instance, by using simple diarylamide derivatives, Li et al. succeeded in functionalizing fullerene in 20–35% yields (calculated on consumed fullerene) [40]. Wang and co-workers exploited arylamide-derived chains carrying a porphyrin moiety to functionalize the C_60_ core in yields up to 44% [41]. Similarly, a diarylamide-based linker was employed in a modified Bingel-type reaction to afford, in 77% yield, the first fullerene-based X-ray contrast agent [42]. Only one example of direct functionalization of C_60_ with a sugar-modified malondiamide-based linker has been previously reported [43]. By exploiting an aliphatic bis-malonamide derivative, Hirsch and coworkers decorated fullerene with fully acetyl-protected α-d-mannopyranose.

Since this last reaction proceeded in a quite low 6% yield, a further goal of our study was to improve the yield of fullerene functionalization with sugar-modified aliphatic diamide chains.

In order to better control the derivatization process, our idea was to assemble the sugar/amino acid-modified malondiamide chains exploiting a copper-mediated Huisgen 1,3-dipolar cycloaddition and then proceed to the C_60_ functionalization.

In the last part of this work, we focused our attention on the synthesis of heterosubstituted C_60_ scaffolds. Actually, a few studies of hetero-functionalization of C_60_ are described in literature [44,45], and there are no examples of C_60_ functionalization with both sugar and amino acid derivatives through the use of malondiamide-based linkers.

## 2. Results

### 2.1. Synthesis of Propargyl-Mannose Glycans

The first step involved the synthesis of natural polymannan analogues. For this purpose mannose-based di- and trisaccharides with α(1,2) and α(1,6) motifs were designed and synthesized. The idea was the obtainment of sugar derivatives simpler than natural polymannans (composed by 10 or more mannose units) and bearing a propargyl group in anomeric position.

The synthesis of the desired di- and trimannans was planned by means of a chemoenzymatic approach. Acetylated building blocks **1a**–**c**, characterized by a free hydroxyl group in position C2 or C6 suitable as acceptors for the synthesis of the corresponding disaccharides **4**–**6**, were prepared by exploiting immobilized enzymes such as *Candida rugosa* lipase (CRL) and *Bacillus pumilus* acetyl xylan esterase (AXE) (Figure 1) [46].

By adopting slightly different reaction conditions (time and temperature), the monodeprotected building blocks (**1a**–**c**) were conjugated by Schmidt glycosylation reaction with man-trichloroacetimidate **2** [47] to give disaccharides **4**–**6** (Figure 2).

Man(α1,6)man disaccharides **4** and **5** were obtained in 86% and 90% yield after 2.5 and 4 h, respectively, working at 0 °C. The synthesis of man(α1,2)man disaccharide (**6**) required lower temperatures (−50/−70 °C) to afford **6** in 80% yield.

The disaccharide **4** was converted into the corresponding trichloroacetimidate **4b** through anomeric deprotection/activation in 52% overall yield. The subsequent coupling reaction with acceptor **1b** afforded, after 4 h at 0 °C, the desired trisaccharide **7** in 56% yield.

### 2.2. Preparation of Glycosylated C_60_ Derivatives

As mentioned above, our approach to fullerene decoration consisted of the use of malondiamide-based linkers previously modified with the desired sugar derivatives through 1,3 Huisgen dipolar cycloaddition. Thus, the synthesis of a malondiamide chain endowed with two terminal azide groups was accomplished (Figure 2). 1-Amino-3-chloropropane hydrochloride was reacted with sodium azide to afford 3-azidopropan-1-amine **8** in a 91% yield. This intermediate was then coupled with malonyl dichloride to give the bis-azido malondiamide **9** in a 44% yield. The click reaction was initially performed between the malondiamide chain **9** and the acetylated monosaccharide **3** to obtain product **10**. Then, the optimized conditions were applied to conjugate disaccharides **5** and **6** and trisaccharide **7**.

The procedure of the Huisgen 1,3-dipolar cycloaddition consisted of dissolving the malondiamide chain in the presence of the proper sugar derivative working in a tetrahydrofuran/H_2_O (1:1) mixture. Then, copper sulphate and sodium ascorbate were added. The desired products were obtained in good yields (78–83%) after flash chromatography purification.

The malondiamide-sugar chains (**10**–**13**) were then reacted with fullerene exploiting a Bingel–Hirsch cyclopropanation for the preparation of the corresponding glycosylated fullerenes (Figure 3).

The mannose-functionalized malondiamide **10** was used as a model compound to optimize the reaction conditions. In a first attempt, **10** was dissolved in dry toluene with C_60_ (1 eq.) and iodine (1.2 eq.), then DBU was added at 0 °C and the mixture was stirred at room temperature for 90 min. After purification, fullerene-man **14** was obtained in a 17% yield in a one-pot fashion (Table 1, entry 1).

Interestingly, monitoring the reaction over time by TLC, we observed the gradual decrease in the product spot and, at the same time, the formation of various new spots ascribable to possible side-products. Assuming that the low conjugation yield could have been due to the degradation of the desired product, the reaction time was shortened. We were pleased to find that functionalized fullerene **14** was obtained in a 30% yield after 1 h (entry 2) and in 41% after 45 min at rt (entry 3).

Each glycosylated malondiamide chain required the application of different reaction conditions to achieve satisfactory yields (Table 1), highlighting a structure–reactivity correlation which characterizes these sugar derivatives. As shown in Table 1, their reactivity decreased as the length of the sugar chain increased. This could be due, probably, to the steric hindrance of the different glycosidic residues, affecting the reactivity of the malondiamide moiety.

The mannosylated chain **11** was first reacted at room temperature for 60 and 90 min to afford fullerene-man(α1,6)man **15** in 12% and 10% yields, respectively (entries 4 and 5). By decreasing the reaction times, the yields increased, and the desired product was isolated in 16% after 30 min and 24 % yields after only 7 min at rt (entries 6 and 8). These results, suggesting a lower stability of **15** at the applied reaction conditions with respect to **14**, prompted us to decrease the reaction temperature. Indeed, by performing the C_60_ functionalization at 0 °C, the yield raised to 31% (entry 11). The same yield was obtained by shortening the reaction time to 1 h (entry 10).

When the reaction conditions adopted for the preparation of fullerene-man(α1,6)man **15** (1 h, 0 °C) were applied to man(α1,2)man malondiamide **12**, the desired product **16** was obtained in a low 8% yield (entry 12). Lowering the temperature to −20 °C did not afford any improvement in terms of yield (entries 13 and 14). On the contrary, when the reaction was performed at rt for three hours, the product was obtained in a 11% yield (entry 15). Longer reaction times resulted again in lower yields (entries 17 and 18). Likewise, the variation in the reagent’s ratio (0.7 eq of sugar and 1 eq of C_60_) did not lead to any improvement.

To obtain fullerene-man(α1,6)man(α1,6)man (**17**) the corresponding malondiamide-chain **13** was initially reacted in the same reaction conditions adopted for the conjugation of man(α1,6)man chain **11**. The desired product **17** was obtained in a 12% yield (entry 19). Attempts to improve the reaction yield by changing reaction time and temperature turned out to be unsuccessful.

### 2.3. Conjugation of C_60_ with Amino Acid Derivatives

By exploiting the approach described for the functionalization of C_60_ with sugar derivatives, the conjugation of a model amino acid such as l-alanine was attempted. The natural amino acid was first converted into the corresponding methyl ester, then its amino group was reacted with 5-hexynoic acid in the presence of EDC*HCl and DMAP to introduce the alkyne moiety (Figure 4). The obtained amino acid derivative **18** was then submitted to azide-alkyne Huisgen cycloaddition with the malondiamide chain by using copper sulphate and sodium ascorbate in a mixture of tetrahydrofuran/water as the solvent [48]. The desired bis-functionalized compound **19** was obtained in a 75% yield after flash chromatography purification (Figure 4).

The first conjugation experiments were performed by following the previously reported protocol based on a Bingel–Hirsch reaction. Given the very poor solubility of the amino acid-chain in toluene, we explored the replacement of toluene with 1,2-dichlorobenzene and the use of various co-solvents, such as DMSO, DMF, DCM, 1,2-dichlorobenzene and CHCl_3_. However, only traces of the desired functionalized product (structure confirmed by HRMS analysis) were obtained. Attempts to improve the yield by prolonging the reaction time, working in the presence of molecular sieves, under different inert atmospheres (argon, nitrogen), or performing the reaction at various temperatures (from –20 to 40 °C) proved to be fruitless; the only difference being, at higher temperatures, the increasing formation of side-products of undefined structures.

These results prompted us to reconsider the functionalization protocol. We decided to preserve the one-pot approach but replacing iodine with CBr_4_ [49]. The addition of DBU to a 1:1 mixture of C_60_ and **19** in toluene, in the presence of CBr_4_, using CHCl_3_ as a co-solvent, allowed the obtainment of the functionalized product **20** in a satisfactory 20% yield.

To prove the applicability of this protocol to different amino acid derivatives such as non-proteinogenic or quaternary amino acids, we decided to prepare the 2-aminoisobutyric (Aib) functionalized malondiamide chain **22**. The preparation and the coupling of the Aib moiety to azide **9** was performed, by following the previously reported protocol, with very similar reaction yields. The same can be said for the conjugation reaction, which afforded the Aib-functionalized fullerene **23** in a 25% yield.

### 2.4. Double Conjugation of C_60_: Sugar and Amino Acid Derivative

With the optimized protocol in our hands, we wondered if it was possible to functionalize C_60_ with both a malondiamide-sugar chain and a malondiamide-amino acid chain in two subsequent steps. Indeed, if proved to be feasible, this protocol would not only be adaptable to the coupling of different derivatives but would also allow controlling the stoichiometry of the reaction in a more accurate fashion.

The mannosylated C_60_ **14** was then reacted with the (Aib)-functionalized malondiamide chain **22** in the presence of CBr_4_ and DBU (Figure 5). Due to the similar solubility of **14** and **22** in CHCl_3_, the reaction was performed in the latter solvent, avoiding the use of toluene. To our delight, TLC monitoring of the reaction revealed the exclusive formation of a new product. After 24 h the reaction mixture was submitted to chromatographic purification, affording the bis-functionalized fullerene **24** in a 37% yield, as a single regioisomer. This result is particularly worthy of note, as, for two different addends, nine regioisomeric bisadducts are in principle possible. Each isomeric bis-addition pattern displayed a typical absorption feature in the visible light region, almost independent of the type of addends and the substituents [50]. Regarding the structural assignment of **24**, the UV-Vis spectrum is in accordance with that of a *trans-3* bisadduct (see Appendix A), which usually is the preferred regioisomer in the case of hetero functionalization with sterically demanding addends [50,51,52]. However, a further confirmation of this data through NMR experiments was hampered due to the broad 1H and 13C signals generated by the presence of rotamers.

In conclusion, the described protocol allows a highly controlled hetero functionalization of Buckminster fullerene, on which the two addends are present in a 1:1 ratio, as confirmed by NMR and HRMS analysis. To the best of our knowledge, this is the first example of the contemporary functionalization of C_60_ with both sugar and amino acid moieties exploiting a malondiamide-based linker.

## 3. Conclusions

We have developed an efficient synthetic protocol for the functionalization of Buckminster fullerene with both polymannan analogues and natural/non-proteinogenic amino acid derivatives.

Simpler analogues of natural polymannan endowed with a propargyl group at the anomeric position were synthesized in their acetylated form exploiting a chemoenzymatic approach. These compounds were submitted to azide-alkyne 1,3 dipolar cycloaddition with a malondiamide-based linker carrying two azido terminal groups. The obtained sugar-modified malondiamide chains were then reacted in the presence of C_60_, affording in a one-pot the corresponding functionalized Buckminster fullerenes with yields ranging from 11% to 41%, higher than those reported in the literature with similar derivatives [43].

Through a slight modification of the functionalization procedure, we demonstrated that natural and non-proteinogenic quaternary amino acids can be also easily linked to C_60_ in a satisfactory 20–26% overall yields.

Finally, the first hetero functionalization of fullerene with both amino acid- and sugar-modified malondiamide linkers has been successfully performed affording the bis-functionalized derivative as a single regioisomer in a 37% yield.

Noteworthy, this conjugation strategy allowed the obtainment of properly decorated fullerenes in a highly controlled fashion.

The derivatization protocol showed to be reliable and, in principle, suitable for the conjugation of more complex oligopeptide derivatives. The synthesis of mannosylated C_60_ endowed with antigenic peptides, which could find future potential application as new generation nanovaccines, is currently under investigation in our group.

## 4. Materials and Methods

Reactants and chemicals were purchased from commercial sources (Sigma-Aldrich, Burlington, MA, USA; Alfa Aesar Ward Hill, MA, USA; Fluorochem Limited, Hadfield, U.K.) and used without further purification. Solvents were purified according to the guidelines in the *Purification of Laboratory Chemicals* [53]. All solvents were freshly distilled from the appropriate drying agent. THF, and toluene were distilled from sodium/benzophenone ketyl; TEA and DCM from CaH_2_. Reactions requiring anhydrous conditions were performed under Ar or N_2_. Yields were calculated for compounds purified by flash chromatography and judged homogeneous by thin-layer chromatography, NMR and mass spectrometry.

Compound purification was performed by flash chromatography using Silica Gel high-purity grade, pore size 60 Å 70–230 mesh, 63–200 μm (Sigma-Aldrich). Analytical thin layer chromatography (TLC) was performed on silica gel F254 precoated aluminium sheets (0.2 mm layer, Merck, Darmstadt, Germany), visualized by a 254 nm UV lamp, and stained with aqueous ceric molybdate solution or iodine and a solution of 4,4′-methylenebis-*N*,*N*-dimethylaniline, ninhydrin, and KI in an aqueous ethanolic solution of AcOH, or by spraying with 5% H_2_SO_4_ in ethanol, followed by heating to 150 °C. Characterization of purified compounds was performed by NMR spectroscopy. NMR spectra were recorded on a Bruker Advance III 400 MHz/600 MHz spectrometer (Bruker Corporation, Billerica, MA, USA). All 1D and 2D NMR spectra were acquired using the standard pulse sequences available with Bruker Topspin 3.6 software package. Chemical shifts (δ) are given in ppm and were referenced to the solvent signals. Signal multiplicities are abbreviated as follows: s, singlet; d, doublet; t, triplet; q, quartet; dd, doublet of doublets; dt, doublet of triplets; td, triplet of doublets; m, multiplet. Structures assignment was performed by means of 2D-COSY and HSQC. Infrared spectra were recorded on a Perkin–Elmer ATR-FTIR 1600 series spectrometer using neat samples. UV-Vis spectra were recorded on a Shimadzu UV-1900 spectrophotometer.

High-resolution mass HRMS data of compounds **7**, **10**–**17**, were acquired using a Bruker Micro-TOF spectrometer in electrospray ionization (ESI) mode, using Tuning-Mix as reference. High-resolution mass HRMS data of compounds **9**, **18**–**24** were acquired using a X500B QTOF System (SCIEX, Framingham, MA 01701 USA) equipped with the Twin Sprayer ESI probe and coupled to an ExionLC™ system (SCIEX). The SCIEX OS software 2.1.6 was used as an operating platform. For MS detection the following parameters were applied: Curtain gas 30 psi, Ion source gas 1 45 psi, Ion source gas 2 55 psi, Temperature 450 °C, Polarity positive, Ionspray voltage 5500 V, TOF mass range 50–2800 Da, declustering potential 60 V and collision energy 10 V.

### 4.1. Sugar Synthesis

The chemoenzymatic synthesis of compounds **1a**–**c**, **2**, **3**, **4**, **4a, 4b**, **5**, and **6** was performed by following previously described procedures, and their spectroscopic data were matched with literature values [46].

Propargyl 2″,3″,4″,6″-tetra-O-acetyl-α-d-mannopyranosyl-(1→6)-2′,3′,4′-tri-O-acetyl-α-d-mannopyranosyl-(1→6)-2,3,4-tri-O-acetyl-α-d-mannopyranoside (**7**)

2′,3′,4′,6′-tetra-O-acetyl-α-d-mannopyranosyl-(1→6)-2,3,4-tri-O-acetyl-α-d-mannopyranosyl trichloracetimidate (**4b**) (250 mg, 0.658 mmol, 2 eq.) and propargyl 2,3,4-tri-O-acetyl-α-d-mannopyranoside (**1b**) (113 mg, 0.329 mmol, 1 eq.) were dissolved in dry dichloromethane (10 mL) in the presence of activated molecular sieves and cooled to 0 °C under nitrogen atmosphere. BF_3_·Et_2_O (40.6 µL, 0.329 mmol, 1 eq.) was added and the mixture was stirred at 0 °C for 4 h. The reaction was quenched with triethylamine (45.644 µL, 0.329 mmol, 1 eq.), stirred for 5 min, filtered and concentrated in vacuo.

The reaction mixture was monitored by TLC (DCM/acetone 9:1, R*_f_* = 0.52 or EtOAc/n-hexane 5:5, R*_f_* = 0.19) and purified by flash chromatography (DCM/acetone 9:1 and EtOAc/n-hexane 5:5). The desired product (181 mg, 56%) was obtain as a white solid.

IR (neat, cm^−1^ 3270, 2956, 2923, 2118, 1741, 1368, 1211, 1038);

^1^H-NMR (400 MHz, CDCl_3_): δ= 2.00, 2.02, 2.07, 2.08, 2.08, 2.13, 2.17, 2.18, 2.20, (9 s, 30H, COCH_3_), 2.53 (t, *J* = 2.4 Hz, 1H, CH_2_C≡CH), 3.55–3.65 (m, 2H), 3.81 (td, *J* = 5 Hz, 11.2 Hz, 2H,), 3.97–4.10 (m, 3H, H-5, H-5′, H5″) 4.15 (dd, *J* = 2.5 Hz, 12.3 Hz, 1H), 4.30 (dd, *J* = 5.6 Hz, 12.3 Hz, 1H), 4.32 (d, *J* = 2.4 Hz, 2H, OC*H*_2_C≡CH), 4.87 (d, *J* = 1.8 Hz, 1H, H-1″), 4.89 (d, *J* = 1.8 Hz, 1H, H-1′) 5.04 (d, *J* = 1.8 Hz, 1H, H-1), 5.28–5.40 (m, 9H, H-2, H-3, H-4, H-2′, H-3′, H-4′, H-2″, H-3″, H-4″);

^13^C{^1^H}-NMR (150 MHz, CDCl_3_): δ= 20.6–20.8 (10C, *C*H_3_COO), 54.8 (O*C*H_2_C≡CH), 62.4, 65.9, 66.3, 66.4, 66.5, 68.6, 68.9, 69.0, 69.2, 69.3, 69.4, 69.9 (15C, ring carbons), 75.6 (OCH_2_C≡*C*H), 78.2 (OCH_2_*C*≡CH), 96.0 (C-1), 97.6, 97.7 (C-1′, C-1″), 169.5–170.5, (10C, CH_3_*C*OO);

HRMS (ESI) calculated for C_41_H_54_O_26_Na [M + Na]^+^ 985.2796, found 985.2796 (Δ = −0.1 ppm).

### 4.2. Functionalization of C_60_ with Sugar Derivatives

3-azidopropan-1-amine (**8**) [54]

3-chloropropylamine hydrochloride (6 g, 46.147 mmol, 1 eq.) and NaN_3_ (11.4 g, 175.358 mmol, 3.8 eq.) was dissolved in H_2_O (20 mL) and the mixture was stirred at 80 °C for 6 h.

The reaction was then treated with saturated KOH solution (10 mL), added dropwise at 0 °C. The aqueous layer was extracted with Et_2_O (3 × 25 mL). The organic phase was dried over MgSO_4,_ filtered and concentrated in vacuo at room temperature (aliquots of pentane were added and evaporated to extract any Et_2_O residue). The desired product was obtained as a yellow oil (4.18 g, 91%). ^1^H and ^13^C NMR were in agreement with previously reported data [54].

*N*^1^,*N*^3^-bis(3-azidopropyl)malondiamide (**9**)

3-azidopropan-1-amine (**8**) (2.5 g, 0.025 mol, 1 eq.) was dissolved in chloroform (20 mL) and a solution of NaOH (2 g) in H_2_O (4 mL) was added. The mixture was cooled to 0 °C and a solution of malonyl chloride (1.21 mL, 0.0125 mol, 0.5 eq.) in chloroform (10 mL) was added dropwise, then was stirred for 20 min at 0 °C. The mixture was extracted with chloroform, the organic phase was dried over MgSO_4_, filtered and concentrated in vacuo.

The reaction mixture was monitored by TLC (ethyl acetate 100%. R*_f_* = 0.45). Column chromatography (SiO_2_, ethyl acetate 100%) gave the desired product (1.5 g, 44%) as a white powder.

IR (neat, cm^−1^) 3287, 3084, 2936, 2870, 2094, 1645, 1626, 1547, 1450, 1423, 1349, 1236, 1197, 725;

^1^H-NMR (400 MHz, CDCl_3_): δ = 1.82 (q, *J* = 6.8 Hz, 4H, NHCH_2_C*H*_2_CH_2_N_3_), 3.20 (s, 2H, COC*H_2_*CO), 3.33–3.42 (m, 8H, NHC*H*_2_CH_2_CH_2_N_3_, NHCH_2_CH_2_C*H*_2_N_3_), 7.47 (t, *J* = 6.4 Hz, 2H, NHCO);

^13^C{^1^H}-NMR (100 MHz, CDCl_3_): δ = 28.5 (NHCH_2_CH_2_CH_2_N_3_), 37.0, 49.1 (2C), 167.6 (*C*OCH_2_*C*O);

MS (ESI) calculated for C_9_H_17_N_8_O_2_ [M + H] ^+^ 269.1469, found 269.1469 (Δ = 0.0 ppm).

Man malondiamide chain (**10**)

Propargyl 2,3,4,6-tetra-O-acetyl-α-d-mannopyranoside (**3**) (86.38 mg, 0.224 mmol, 3 eq.) was dissolved in THF (8 mL) and H_2_O (8 mL). *N*^1^,*N*^3^-bis(3-azidopropyl)malondiamide (**9**) (20 mg, 0.0746 mmol, 1 eq.) then copper sulphate pentahydrate (186.2 mg, 0.746 mmol, 10 eq.) and sodium ascorbate (252 mg, 1.268 mmol, 17 eq.) were added and the mixture was stirred overnight at rt. The reaction was extracted with dichloromethane, and the organic layer was washed with saturated NaHCO_3_ solution, then dried over MgSO_4_, filtered and concentrated in vacuo.

The reaction mixture was analysed by TLC (DCM/MeOH 95:5, R*_f_* = 0.31). Column chromatography (SiO_2_, dichloromethane/MeOH 95:5) gave the desired product (62 mg, 80%) as white solid.

IR (neat, cm^−1^) 3142, 3095, 2921, 2851, 1741, 1653, 1435, 1368, 1216, 1078, 1044;

^1^H-NMR (400 MHz, CDCl_3_): δ = 2.00, 2.06, 2.14, 2.17 (4 s, 24H, COCH_3_), 2.18–2.25 (m, 4H, NHCH_2_C*H*_2_CH_2_N), 3.16 (s, 2H, COC*H*_2_CO), 3.35 (s, 4H, NHC*H*_2_CH_2_CH_2_N), 4.31 (dd, *J* = 4.9 Hz, 12.2 Hz, 2H, H-6a), 4.47 (t, *J* = 5.6 Hz, 4H, NHCH_2_CH_2_C*H*_2_N), 4.71 (d, *J* = 11.5 Hz, 2H, triazole-CH_2_O), 4.87 (d, *J* = 11.5 Hz, 2H, triazole-CH_2_O), 5.25 (s, 2H, H-2), 4.99 (s, 2H, H-1), 4.06–4.17 (m, 4H, H-5, H-6b), 5.33 (m, 4H, H-3, H-4), 7.18 (s, 2H, Ar C*H* triazole), 7.74 (s, 2H, CON*H*CH_2_);

^13^C{^1^H}-NMR (100 MHz, CDCl_3_): δ = 20.6–20.8 (8C, *C*H_3_COO), 29.9 (NHCH_2_*C*H_2_CH_2_N), 36.5 (NH*C*H_2_CH_2_CH_2_N), 42.8 (CO*C*H_2_CO), 47.9 (NHCH_2_CH_2_*C*H_2_N), 60.9 (triazole-CH_2_O), 62.4 (C-6a, C-6b), 66.0 (C-3), 68.7 (C-5), 69.1 (C-2), 69.4 (C-4), 96.8 (C-1), 123.6 (N*C*H=C), 143.5, 167.8 (*C*OCH_2_*C*O), 169.7–170.7 (8C, CH_3_*C*OO);

HRMS (ESI) calculated for C_43_H_60_N_8_O_22_Na [M + Na] ^+^ 1063.3714, found 1063.3716 (Δ = −0.2 ppm).

Man(α1,6)man malondiamide chain (**11**)

Propargyl-2′,3′,4′,6′-tetra-O-acetyl-α-d-mannopyranosyl-(1→6)-2,3,4-tri-O-acetyl-α-d-mannopyranoside (**5**) (37.6 mg, 0.0558 mmol, 3 eq.) was dissolved in THF (2.5 mL) and H_2_O (2.5 mL). *N*^1^,*N*^3^-bis(3-azidopropyl) malondiamide (**9**) (5 mg, 0.0186 mmol, 1 eq.) then copper sulphate pentahydrate (46.4 mg, 0.186 mmol, 10 eq.) and sodium ascorbate (63 mg, 0.316 mmol, 17 eq.) were added and the mixture was stirred overnight at rt. The reaction was extracted with dichloromethane, and the organic layer was washed with saturated NaHCO_3_ solution, then dried over MgSO_4_, filtered and concentrated in vacuo. The reaction mixture was analysed by TLC (DCM/MeOH 95:5, R*_f_* = 0.28). Column chromatography (SiO_2_, dichloromethane/MeOH 95:5) gave the desired product (25 mg, 83%) as a white solid.

IR (neat, cm^−1^) 3142, 3080, 2922, 2852, 1744, 1670, 1462, 1368, 1218, 1080, 1041;

^1^H-NMR (600 MHz, CDCl_3_): δ = 1.90, 1.91, 1.97, 1.98, 2.04, 2.08, 2.09 (7 s, 42H, COCH_3_), 2.10–2.16 (m, 4H, NHCH_2_C*H*_2_CH_2_N), 3.07 (s, 2H, COC*H_2_*CO), 3.25 (s, 4H, NHC*H*_2_CH_2_CH_2_N), 3.54 (dd, *J* = 2.2 Hz, 11.1 Hz, 2H, H-6b), 3.74 (dd, *J* = 5.3 Hz, 11.0 Hz, 2H, H-6a), 3.96–4.01 (m, 2H, H-5), 4.04 (ddd, *J* = 2.4 Hz, 5.2 Hz, 9.7 Hz, 2H, H-5′), 4.08 (dd, *J* = 2.4 Hz, 12.2 Hz, 4H, H-6′b), 4.21 (dd, *J* = 5.3 Hz, 12.2 Hz, 2H, H-6′a), 4.39 (s, 4H, NHCH_2_CH_2_C*H*_2_N), 4.61 (s, 2H, triazole-C*H_2_*O), 4.77 (s, 2H, triazole-C*H_2_*O), 4.83 (d, *J* = 1.7 Hz, 2H, H-1′), 4.87 (s, 2H, H-1), 5.15–5.29 (m, 12H, H-2, H-2′, H-3, H-3′, H-4, H4′), 7.23 (s, 2H, CON*H*CH_2_), 7.74 (s, 2H, Ar C*H* triazole);

^13^C{^1^H}-NMR (150 MHz, CDCl_3_): δ = 20.6–20.8 (14C, *C*H_3_COO), 29.7 (NHCH_2_*C*H_2_CH_2_N), 36.6 (NH*C*H_2_CH_2_CH_2_N), 42.6 (CO*C*H_2_CO), 47.9 (NHCH_2_CH_2_*C*H_2_N), 60.6 (triazole-CH_2_O), 62.4, 65.9 (C-6′a, C-6′b), 66.5 (C-6a, C-6b), 66.6, 68.7, 69.1, 69.3, 69.4, 69.5, 96.6 (C-1), 97.5 (C-1′), 167.6, (*C*OCH_2_*C*O), 169.7–170.6 (14C, CH_3_*C*OO);

HRMS (ESI) calculated for C_67_H_92_N_8_O_38_Na [M + Na] ^+^ 1639.5405, found 1639.5405 (Δ = 0.0 ppm).

Man(α1,2)man malondiamide chain (**12**)

Propargyl-2′,3′,4′,6′-tetra-O-acetyl-α-d-mannopyranosyl-(1→2)-3,4,6-tri-O-acetyl-α-D-mannopyranoside (**6**) (71 mg, 0.105 mmol, 3 eq) was dissolved in THF (3 mL) and H_2_O (3 mL). *N*^1^,*N*^3^-bis(3-azidopropyl) malondiamide (**9**) (9.4 mg, 0.0351 mmol, 1 eq.) then copper sulphate pentahydrate (118.8 mg, 0.351 mmol, 10 eq.) and sodium ascorbate (87.64 mg, 0.596 mmol, 17 eq.) were added and the mixture was stirred overnight at rt. The reaction was extracted with dichloromethane, and the organic layer was washed with saturated NaHCO_3_ solution, then dried over MgSO_4_, filtered and concentrated in vacuo.

The reaction mixture was monitored by TLC (dichloromethane/MeOH 95:5, R*_f_* = 0.28). Column chromatography (SiO_2_, dichloromethane/MeOH 95:5) gave the desired product (43.7 mg, 77%) as white solid.

IR (neat, cm^−1^) 3142, 3084, 2919, 2850, 1748, 1669, 1463, 1368, 1222, 1084, 1041;

^1^H-NMR (400 MHz, CDCl_3_): δ = 2.01, 2.02, 2.05, 2.07, 2.10, 2.14, 2.15, (7 s, 42H, COCH_3_), 2.16–2.23 (m, 4H, NHCH_2_C*H*_2_CH_2_N), 3.18 (s, 2H, CO*CH_2_*CO), 3.32 (s, 4H, NHC*H*_2_CH_2_CH_2_N), 3.96–4.27 (m, 14H, H-2, H-5a, H5′b, H-6a, H-6b, H-6′a, H-6′b,), 4.46 (s, 4H, NHCH_2_CH_2_C*H*_2_N), 4.68 (s, 2H, triazole-CH_2_O), 4.83 (s, 2H, triazole-CH_2_O), 4.92 (s, 2H, H-1′), 5.14 (s, 2H, H-1), 5.23–5.43 (m, 10H, H-2′, H-3, H-3′, H-4, H-4′), 7.37 (s, 2H, CON*H*CH_2_), 7.75 (s, 2H, Ar C*H* triazole);

^13^C{^1^H}-NMR (100 MHz, CDCl_3_): δ = 20.6–20.8 (14C, *C*H_3_COO), 29.8 (NHCH_2_*C*H_2_CH_2_N), 36.5 (NH*C*H_2_CH_2_CH_2_N), 42.6 (CO*C*H_2_CO), 47.8 (NHCH_2_CH_2_*C*H_2_N), 53.4, 60.9 (triazole-CH_2_O), 62.1, 62.4, 66.1, 66.3, 68.3, 68.7, 69.1, 69.7, 70.2, 97.7 (C-1), 99.1 (C-1′), 167.7, (*C*OCH_2_*C*O), 169.3–170.9 (14C, CH_3_*C*OO);

HRMS (ESI) calculated for C_67_H_92_N_8_O_38_Na [M + Na] ^+^ 1639.5405, found 1639.5405 (Δ = 0.0 ppm).

Man(α1,6)man(α1,6)man malondiamide chain (**13**)

Propargyl 2″,3″,4″,6″-tetra-O-acetyl-α-d-mannopyranosyl-(1→6)-2′,3′,4′-tri-O-acetyl-α-d-mannopyranosyl-(1→6)-2,3,4-tri-O-acetyl-α-d-mannopyranoside (**7**) (32 mg, 0.0336 mmol, 3 eq.) was dissolved in THF (2 mL) and H_2_O (2 mL). *N*^1^,*N*^3^-bis(3-azidopropyl) malondiamide (**9**) (3 mg, 0.0112 mmol, 1 eq.) then copper sulphate pentahydrate (27.9 mg, 0.112 mmol, 10 eq.) and sodium ascorbate (37.9 mg, 0.190 mmol, 17 eq.) were added and the mixture was stirred overnight at rt. The reaction was extracted with dichloromethane, and the organic layer was washed with saturated NaHCO_3_ solution, then dried over MgSO_4_, filtered and concentrated in vacuo. The reaction mixture was monitored by TLC (DCM/MeOH 95:5, R*_f_* = 0.30). Column chromatography (SiO_2_, DCM/MeOH 95:5) gave the desired product (18.5 mg, 75%) as white solid.

IR (neat, cm^−1^) 3142, 3080, 2920, 2850, 1743, 1671, 1484, 1368, 1216, 1081, 1040;

^1^H-NMR (600 MHz, CDCl_3_): δ = 1.90, 1.91, 1.98, 1.99, 1.99, 2.05, 2.08, 2.09, 2.10, (9 s, 60H, COCH_3_), 2.11–2.15 (m, 4H, NHCH_2_C*H*_2_CH_2_N), 3.06 (s, 2H, COCH_2_CO), 3.19–3.31 (m, 4H, NHC*H*_2_CH_2_CH_2_N), 3.50–3.56 (m, 4H), 3.73 (dd, *J* = 3.3Hz, 7.5 Hz, 2H), 3.77 (dd, *J* = 3.3Hz, 7.5 Hz, 2H,), 3.93–4.02 (m, 6H, H-5, H-5′, H-5″), 4.06 (dd, J =1.5 Hz, 8.3 Hz, 2H), 4.21 (dd, *J* = 3.6 Hz, 8.3 Hz, 2H), 4.38 (t, 4H, *J* = 4.3 Hz, NHCH_2_CH_2_C*H*_2_N), 4.62 (s, 2H, triazole-C*H_2_*O), 4.77 (s, 2H, triazole-CH_2_O), 4.81 (s, 4H, H-1′, H-1″), 4.88 (s, 2H, H-1), 5.16 (s, 2H, H-2), 5.19–5.32 (m, 16H, H-2′, H-2″, H-3, H-3′, H-3″, H-4, H-4′, H-4″), 7.18 (s, 2H, CON*H*CH_2_), 7.70 (s, 2H, Ar C*H* triazole);

^13^C{^1^H}-NMR (150 MHz, CDCl_3_): δ = 20.6–20.8 (20C, *C*H_3_COO), 29.6 (NHCH_2_*C*H_2_CH_2_N), 29.7, 36.6 (NH*C*H_2_CH_2_CH_2_N), 42.6 (CO*C*H_2_CO), 47.8 (NHCH_2_CH_2_*C*H_2_N), 60.7 (triazole-CH_2_O), 62.4, 65.9, 66.2, 66.3, 66.4, 66.5, 68.6, 69.0, 69.2, 69.3, 69.4, 69.6, 96.5 (C-1), 97.6, 97.7 (C-1′, C.1″), 167.7 (*C*OCH_2_*C*O), 169.6–170.6 (20C, CH_3_*C*OO);

HRMS (ESI) calculated for C_91_H_125_N_8_O_54_Na [M + Na] ^+^ 2216.7128, found 2216.7143 (Δ = +1.4 ppm).

Fullerene-man (**14**)

C_60_ (7 mg, 0.0096 mmol, 1 eq.), compound **10** (10 mg, 0.0096 mmol, 1 eq.) and I_2_ (3 mg, 0.0117 mmol, 1.22 eq.) were dissolved in dry toluene (4.5 mL). DBU (3 μL, 0.0216 mmol, 2.25 eq.) was added to the mixture under argon at 0 °C and the reaction was stirred at rt for 45 min. The reaction mixture was monitored by TLC (DCM/acetone 1:1. R*_f_* = 0.77). The solvent was removed under reduced pressure and a column chromatography (SiO_2_, DCM/acetone 1:1) gave the desired product (7 mg, 41%) as a black-brown solid.

IR (neat, cm^−1^) 3267, 3142, 3074, 2919, 2850, 1746, 1653, 1428, 1367, 1218, 1079, 1045;

^1^H-NMR (400 MHz, CDCl_3_): δ = 2.00, 2.06, 2.14, 2.17 (4 s, 24H, COCH_3_), 3.54 (s, 4H, NHC*H*_2_CH_2_CH_2_N), 2.35–2.55 (m, 4H, NHCH_2_C*H*_2_CH_2_N), 4.10 (s, 2H, H-5), 4.28 (m, 4H, H-6a, H-6b), 4.61–4.89 (m, 8H, triazole-CH_2_O, NHCH_2_CH_2_C*H*_2_N), 4.98 (s, 2H, H-1), 5.27 (s, 2H, H-2), 5.30–5.39 (m, 4H, H-3, H-4), 7.85 (s, 2H, CON*H*CH_2_), 9.00 (s, 2H, Ar C*H* triazole);

^13^C{^1^H}-NMR (150 MHz, CDCl_3_): δ = 20.7–29.3 (8C), 29.7, 31.7, 31.9 37.2, 47.8, 53.8, 60.9, 62.4, 65.9, 69.0, 69.2, 69.5, 96.6, 123.6, 136.5–146.1 (C-C_60_), 163.8–170.3 (8C);

HRMS (ESI) calculated for C_103_H_58_N_8_O_22_Na [M + Na] ^+^ 1782.3590, found 1782.3555 (Δ = +1.8 ppm).

Fullerene-man(α1,6)man (**15**)

C_60_ (6.24 mg, 0.00866 mmol, 1 eq.), Compound **11** (14 mg, 0.00866 mmol, 1 eq.) and I_2_ (2.68 mg, 0.0105 mmol, 1.22 eq.) were dissolved in dry toluene (4.5ml). DBU (2.9 μL, 0.0195 mmol, 2.25 eq.) was added to the solution under argon at 0 °C and the reaction was stirred at 0 °C for 1 h. The reaction mixture was monitored by TLC (DCM/Acetone 6:4, R*_f_* = 0.75). The solvent was removed under reduced pressure and a column chromatography (SiO_2_, DCM/acetone 6:4) gave the desired product (6 mg, 30 %) as a black-brown solid.

IR (neat, cm^−1^) 3267, 3147, 3074, 2921, 2851, 1746, 1654, 1429, 1367, 1218, 1082, 1042;

^1^H-NMR (600 MHz, CDCl_3_): δ = 1.90, 1.96, 1.97, 2.04, 2.08 (5s, 42H, COCH_3_), 2.24–2.34 (m, 4H, NHCH_2_C*H*_2_CH_2_N), 3.50 (t, *J* = 6.7 Hz, 4H, NHC*H*_2_CH_2_CH_2_N), 3.53 (dd, *J* = 2.6 Hz, 11.1 Hz, 2H, H-6b), 3.74 (dd, *J* = 5.8 Hz, 11.1 Hz, 2H, H-6a), 3.96–4.00 (m, 2H, H-5), 4.09 (dd, *J* = 2.4 Hz, 12.3 Hz, 2H, H-6′b), 4.20 (dd, *J* = 5.3 Hz, 12.2 Hz, 2H, H-6′a), 4.50 (t, *J* = 6.6 Hz, 4H, NHCH_2_CH_2_C*H*_2_N), 4.61–4.79 (m, 4H, triazole-CH_2_O), 4.81 (d, *J* = 2.7 Hz, 2H, H-1′), 4.86 (s, 2H, H-1), 5.15–5.29 (m, 12 H, H-2, H-2′, H-3, H-3′, H-4, H-4′), 7.22 (s, 2H, Ar C*H* triazole), 7.88 (s, 2H, CON*H*CH_2_);

^13^C{^1^H}-NMR (150 MHz, CDCl_3_): δ = 20.7, 20.8, 20.9, 29.2–29.6, 29.7, 31.9, 33.4, 37.6, 47.5, 53.8, 60.6, 62.5, 65.9, 66.5, 66.6, 68.7, 69.1, 69.2, 69.3, 69.5, 69.6, 73.7, 96.4, 97.5, 123.7, 137.7–145.3 (C- C_60_), 163.4, 169.7, 170.0, 170.3, 170.7;

HRMS (ESI) calculated for C_127_H_90_N_8_O_38_Na [M + Na] ^+^ 2358.5281, found 2358.5283 (Δ = −1.0 ppm).

Fullerene-man(α1,2)man (**16**)

C_60_ (4.4 mg, 0.00618 mmol, 1 eq.), Compound **12** (10 mg, 0.00618 mmol, 1 eq.) and I_2_ (1.9 mg, 0.00754 mmol, 1.22 eq.) were dissolved in dry toluene (4 mL). DBU (2.1 μL, 0.0139 mmol, 2.25 eq.) was added to the solution under argon at 0 °C and the reaction was stirred at rt for 1.5 h. The reaction mixture was monitored by TLC (DCM/acetone 6:4, R*_f_* = 0.73). The solvent was removed under reduced pressure and a column chromatography (SiO_2_, DCM/acetone 6:4) gave the desired product (1.4 mg, 10%) as a black-brown solid.

IR (neat, cm^−1^) 3267, 3136, 3073, 2921, 2851, 1743, 1654, 1429, 1367, 1221, 1081, 1044;

^1^H-NMR (400 MHz, CDCl_3_): δ = 2.03, 2.04, 2.07, 2.09, 2.13, 2.17, 2.18, (7s, 42H, COCH_3_), 2.35–2.38 (m, 4H, NHCH_2_C*H*_2_CH_2_N), 3.61 (m, 4H, NHC*H_2_*CH_2_CH_2_N), 3.99–4.27 (m, 14H, H-2, H-5, H5′b, H-6a, H-6b, H-6′a, H-6′b,), 4.56 (m, 4H, NHCH_2_CH_2_C*H_2_*N), 4.73 (d, 2H, *J* = 12.3 Hz, triazole-CH_2_O), 4.87 (d, 2H, *J* = 12.3 Hz, triazole-CH_2_O), 4.94 (d, *J* = 1.9 Hz, 2H, H-1′), 5.15 (s, 2H, H-1), 5.25–5.44 (m, 10H, H-2′, H-3, H-3′, H-4, H4′), 7.74 (s, 2H, CON*H*CH_2_), 8.00 (s, 2H, Ar C*H* triazole);

^13^C{^1^H}-NMR (150 MHz, CDCl_3_): δ = 20.9, 20.8, 20.7, 29.6–29.2, 29.7, 31.9, 62.3, 62.7, 66.1, 66.5, 69.6, 70.2, 137.7–145.3 (C- C_60_), 169.7, 170.95;

HRMS (ESI) calculated for C_127_H_90_N_8_O_38_Na [M + Na] ^+^ 2358.5281, found 2358.5225 (Δ = +3.2 ppm).

Fullerene-man(α1,6)man(α1,6)man (**17**)

C_60_ (5 mg, 0.007 mmol, 1 eq.), Compound **13** (15.5 mg, 0.007 mmol, 1 eq.) and I_2_ (2.18 mg, 0.00854 mmol, 1.22 eq.) were dissolved in dry toluene (4 mL). DBU (2.4 μL, 0.0157 mmol, 2.25 eq.) was added to the mixture under argon at 0 °C and the reaction was stirred at rt for 1.5 h. The reaction mixture was monitored by TLC (DCM/acetone 6:4, R*_f_* = 0.74). The solvent was removed and a column chromatography (SiO_2_, DCM/acetone 6:4) gave the desired product (3 mg, 10%) as a black-brown solid.

IR (neat, cm^−1^) 3268, 3134, 3073, 2922, 2851, 1744, 1653, 1430, 1367, 1214, 1083, 1041;

^1^H-NMR (400 MHz, CDCl_3_): δ = 1.99, 2.07, 2.08, 2.13, 2.16, 2.17, 2.18 (7s, 60H, COCH_3_). 2.34–2.47 (m, 4H, NHC*H*_2_CH_2_CH_2_N), 3.61 (d, *J* = 10.5 Hz, 6H), 3.83 (ddd, *J* = 6.2 Hz, 11.2 Hz, 15.9 Hz, 4H,), 3.96–4.21 (m, 10H), 4.30 (dd, *J* = 5.2 Hz, 12.3 Hz, 4H), 4.56–5.07 (m, 12H), 5.21–5.50 (m, 18H), 8.50 (s, 2H, Ar C*H* triazole);

^13^C{^1^H}-NMR (150 MHz, CDCl_3_): δ = 20.6–20.9 (30C), 29.2, 29.7, 31.9, 32.5, 37.6, 47.7, 53.8, 60.3, 62.4, 65.9, 66.2, 66.3, 66.5, 68.6, 69.1, 69.3, 69.4, 69.5, 69.8, 96.6, 97.6, 97.8, 137.6–145.6 (C-C_60_), 163.4, 169.6–170.6 (20 C);

HRMS (ESI) calculated for C_151_H_122_N_8_O_54_Na [M + Na] ^+^ 2934.6971, found 2934.6875 (Δ = +1.9 ppm).

### 4.3. Functionalization of C_60_ with Amino Acid Derivatives

Methyl hex-5-ynoyl-l-alaninate (**18**)

A suspension of l-alanine methyl ester hydrochloride salt (125 mg, 0.896 mmol, 1 eq.) in DCM (0.1 M) was added with 5-hexynoic acid (95 μL, 0.896 mmol, 1 eq.) and DMAP (196 mg, 1.60 mmol, 1.8 eq.) at rt under nitrogen atmosphere. The mixture was cooled at 0 °C and EDC·HCl (172 mg, 0.896 mmol, 1 eq.) was added. After 30 min the reaction was allowed to warm to rt and maintained under stirring overnight.

The reaction was added with 15 mL of 1 M HCl and extracted with DCM. The organic phases were collected and washed with saturated NaHCO_3_ solution, then dried over Na_2_SO_4_, filtered, and concentrated in vacuo to give **18** (155 mg, 89%) as a colourless oil without the need of further purifications.

The reaction was monitored by TLC analysis (EtOAc/Hex 6:4, R*_f_* = 0.33).

IR (neat, cm^−1^) 3287, 3063, 2952, 2840, 2116, 1740, 1643, 1534, 1453, 1379, 1207, 1165, 1061;

^1^H-NMR (CDCl_3_, 400 MHz,) δ 1.40 (d, *J* = 7.2 Hz, 3H, CHC*H_3_*), 1.86 (quintuplet, *J* = 7.0 Hz, 2H, CH_2_C*H_2_*CH_2_), 1.98 (t, *J* = 2.6 Hz, 1H, C≡C*H*), 2.26 (td, *J* = 2.6, 7.0 Hz, 2H, C*H_2_*C≡CH), 2.36 (t, *J* = 7.1 Hz, 2H, COC*H_2_*), 3.74 (s, 3H, CO_2_*Me*), 4.49 (quintuplet, *J* = 7.2 Hz, 1H, C*H*CH_3_), 6.21 (br d, *J* = 7.0 Hz, 1H, N*H*);

^13^C{^1^H}-NMR (CDCl_3_, 100 MHz) δ 17.1 (t, *C*H_2_CCH), 18.4 (d, CH*C*H_3_), 24.0 (t, CH_2_*C*H_2_CH_2_), 34.8 (t, CO*C*H2), 47.9 (d, *C*HCH_3_), 52.4 (q, CO_2_*Me*), 69.2 (d, C≡*C*H), 83.4 (s, *C*≡CH), 171.8 (s, *C*ONH), 173.6 (s, *C*O_2_Me);

HRMS (ESI) calculated for C_10_H_16_NO_3_ [M + H] ^+^ 198.1125, found 198.1123 (Δ = −0.9 ppm).

Dimethyl 2,2′-((4,4′-(((malonylbis(azanediyl))bis(propane-3,1-diyl))bis(1*H*-1,2,3-triazole-1,4-diyl))bis(butanoyl))bis(azanediyl))dipropionate (**19**)

Compound **18** (100 mg, 0.581 mmol, 2.1 eq.) and *N*^1^,*N*^3^-bis(3-azidopropyl)malondiamide **9** (75 mg, 0.227 mmol, 1 eq.) were dissolved in 27 mL of THF/H_2_O (2:1) mixture and added with copper sulphate pentahydrate (48 mg, 0.193 mmol, 0.7 eq.) and sodium ascorbate (61 mg, 0.304 mmol, 1.1 eq.). Reaction completion was monitored by TLC analysis (EtOAc/Hex 6:4 to evaluate starting material consumption; DCM/MeOH 8:2 to monitor product formation) and, after 4 h, the solvent was evaporated at reduced pressure. The crude mixture was purified by flash chromatography (DCM/MeOH 85:15) to obtain compound **19** as a white solid (135 mg, 73%) R*_f_* = 0.29.

IR (neat, cm^−1^) 3298, 3120, 3066, 2952, 2850, 1736, 1638, 1625, 1539, 1211, 1166, 1055;

^1^H-NMR (CDCl_3_, 400 MHz,) δ 1.41 (d, *J* = 7.2 Hz, 6H, CHC*H_3_*), 2.01 (quintuplet, *J* = 7.1 Hz, 4H, COCH_2_C*H_2_*CH_2_), 2.16 (m, 4H, NHCH_2_C*H_2_*CH_2_N), 2.27 (t, *J* = 7.3 Hz, 4H, COC*H_2_*CH_2_CH_2_), 2.79 (t, *J* =7.0, 4H, COCH_2_CH_2_C*H_2_*), 3.17 (s, 2H, COC*H_2_*CO), 3.27 (q, *J* = 6.2 Hz, 4H, NHC*H_2_*CH_2_CH*_2_*N), 3.76 (s, 6H, CO_2_*Me*), 4.41 (t, *J* = 6.5, 4H, NHCH_2_CH_2_*CH_2_*), 4.60 (q, *J* = 7.2, 2H, C*H*CH_3_), 6.91 (br d, *J* = 7.3, 2H, CON*H*CHCH_3_), 7.44 (s, 2H, CON*H*CH_2_), 7.46 (s, 2H, Ar C*H* triazole);

^13^C{^1^H}-NMR (CDCl_3_, 100 MHz) δ 18.1, 24.4, 25.2, 29.7, 34.9, 36.5, 43.0, 47.5, 48.0, 52.4, 121.8, 147.2, 167.7, 172.5, 173.9;

HRMS (ESI) calculated for C_29_H_47_N_10_O_8_Na [M + H] ^+^ 663.3573, found 663.3563 (Δ = −1.5 ppm).

Fullerene-l-alanine (**20**)

C_60_ (55 mg, 0.075 mmol, 1 eq.), compound **19** (50 mg, 0.075 mmol, 1 eq.) and CBr_4_ (28 mg, 0.083 mmol, 1.1 eq.) were dissolved in 28 mL of toluene/CHCl_3_ (3:1). DBU (25 μL, 0.166 mmol, 2.2 eq.) was added to the mixture under nitrogen atmosphere at 0 °C and the reaction was allowed to warm to rt. After 48 h the solvent was evaporated at reduced pression and the crude mixture was purified by flash chromatography (toluene to recover unreacted C_60_, then DCM/MeOH 9:1) to obtain compound **20** as a red-brown solid (21mg, 20%) R*_f_* = 0.38.

IR (neat, cm^−1^) 3278, 3075, 2929, 2874, 1738, 1655, 1534, 1452, 1213, 1023, 1002;

^1^H-NMR (DMSO-*d*_6_, 400 MHz,) δ 1.26 (d, *J* = 7.3 Hz, 6H, CHC*H_3_*), 1.81 (quintuplet, *J* = 7.3 Hz, 4H, COCH_2_C*H_2_*CH_2_), 2.09–2.20 (m, 8H, NHCH_2_C*H_2_*CH_2_N + COC*H_2_*CH_2_CH_2_), 2.61 (t, *J* = 7.4 Hz, 4H, COCH_2_CH_2_C*H_2_*), 3.34–3.42 (m, 4H, NHC*H_2_*CH_2_CH*_2_*N), 3.61 (s, 6H, CO_2_*Me*), 4.25 (quintuplet, *J* = 7.3 Hz, 2H, C*H*CH_3_), 4.40 (t, *J* = 7.0, 4H, NHCH_2_CH_2_*CH_2_*), 7.88 (s, 2H, Ar C*H* triazole), 8.25 (d, *J* = 7.0 Hz, 2H, CON*H*CHCH_3_), 9.59 (t, *J* = 5.6 Hz, 2H, CON*H*CH_2_);

^13^C{^1^H}-NMR (DMSO-*d*_6_, 100 MHz) δ 17.4, 25.0, 25.6, 29.5, 30.2, 34.8, 37.6, 47.9, 52.2, 55.4, 122.4, 148.0, 137.2–146.9 (C-C_60_), 162.4, 172.3, 173.7;

HRMS (ESI) calculated for C_89_H_45_N_10_O_8_ [M + H] ^+^ 1381.3416, found 1381.3429 (Δ = +0.9 ppm).

Methyl 2-(hex-5-ynamido)-2-methylpropanoate (**21**)

A suspension of 2-aminoisobutyric acid methyl ester hydrochloride salt (145 mg, 0.945 mmol, 1 eq.) in DCM (0.1 M) was added with 5-hexynoic acid (100 μL, 0.945 mmol, 1 eq.) and DMAP (208 mg, 1.70 mmol, 1.8 eq.) at rt under nitrogen atmosphere. The mixture was cooled at 0 °C and EDC·HCl (181 mg, 0.945 mmol, 1 eq.) was added. After 30 min, the reaction was allowed to warm to rt and maintained under stirring overnight.

The reaction was added with 15 mL of 1 M HCl and extracted with DCM. The organic phases were collected and washed with saturated NaHCO_3_ solution, then dried over Na_2_SO_4_, filtered, and concentrated in vacuo to give **21** (161 mg, 81%) as a white solid without the need of further purifications.

The reaction was monitored by TLC analysis (EtOAc/Hex 6:4, R*_f_* = 0.39).

IR (neat, cm^−1^) 3275, 3058, 2932, 2118, 1731, 1636, 1539, 1154;

^1^H-NMR (CDCl_3_, 400 MHz,) δ 1.51 (s, 6H, C(C*H_3_*)_2_), 1.82 (quintuplet, *J* = 7.1 Hz, 2H, COCH_2_C*H_2_*CH_2_), 1.96 (t, *J* = 2.6 Hz, 1H, C≡C*H*), 2.24 (td, *J* = 2.6, 7.0 Hz, 2H, C*H_2_*C≡CH), 2.29 (t, *J* = 7.2 Hz, 2H, COC*H_2_*), 3.71 (s, 3H, CO_2_*Me*), 6.14 (br s, 1H, N*H*);

^13^C{^1^H}-NMR (CDCl_3_, 100 MHz) δ δ 17.6 (t, *C*H_2_CCH), 24.0 (t, CH_2_*C*H_2_CH_2_), 24.9 (q, C(*C*H_3_)_2_), 35.0 (t, CO*C*H2), 52.6 (q, CO_2_*Me*), 56.4 (s, *C*(CH_3_)_2_), 69.1 (d, C≡*C*H), 83.6 (s, *C*≡CH), 171.6 (s, *C*ONH), 175.0 (s, *C*O_2_Me);

HRMS (ESI) calculated for C_11_H_18_NO_3_ [M + H]^+^ 212.1281, found 212.1279 (Δ = −1.0 ppm).

Dimethyl 2,2′-((4,4′-(((malonylbis(azanediyl))bis(propane-3,1-diyl))bis(1*H*-1,2,3-triazole-1,4-diyl))bis(butanoyl))bis(azanediyl))bis(2-methylpropanoate) (**22**)

Compound **21** (101 mg, 0.478 mmol, 2.1 eq.) and *N*^1^,*N*^3^-bis(3-azidopropyl)malondiamide **9** (61 mg, 0.227 mmol, 1 eq.) were dissolved in 21 mL of THF/H_2_O (2:1) mixture and added with copper sulphate pentahydrate (39 mg, 0.158 mmol, 0.7 eq.) and sodium ascorbate (49 mg, 0.248 mmol, 1.1 eq.). Reaction completion was monitored by TLC analysis (EtOAc/Hex 6:4 to evaluate starting material consumption; DCM/MeOH 8:2 to monitor product formation) and, after 3 h, the solvent was evaporated at reduced pressure. The crude mixture was purified by flash chromatography (DCM/MeOH 80:20) to obtain compound **22** as a colourless oil (152 mg, 98%) R*_f_* = 0.37.

IR (neat, cm^−1^) 3278, 3065, 2948, 1737, 1642, 1534, 1149;

^1^H-NMR (CDCl_3_, 400 MHz,) δ 1.52 (s, *12*H, C(C*H_3_*)_2_), 1.97 (quintuplet, *J* = 7.1 Hz, 4H, COCH_2_C*H_2_*CH_2_), 2.15 (quintuplet, *J* = 6.5 Hz, 4H, NHCH_2_C*H_2_*CH_2_N), 2.20 (t, *J* = 7.1 Hz, 4H, COC*H_2_*CH_2_CH_2_), 2.77 (t, *J* = 7.0, 4H, COCH_2_CH_2_C*H_2_*), 3.13 (s, 2H, COC*H_2_*CO), 3.28 (q, *J* = 6.3 Hz, 4H, NHC*H_2_*CH_2_CH*_2_*N), 3.73 (s, 6H, CO_2_*Me*), 4.40 (t, *J* = 6.5, 4H, NHCH_2_CH_2_*CH_2_*), 6.77 (s, 2H, CON*H*C(CH_3_)_2_), 7.27 (br s, 2H, CON*H*CH_2_), 7.45 (s, 2H, Ar C*H* triazole);

^13^C{^1^H}-NMR (CDCl_3_, 100 MHz) δ 24.3, 25.0, 25.3, 29.8, 35.0, 36.5, 42.7, 47.6, 52.5, 56.1, 122.0, 147.2, 168.0, 172.4, 175.2;

HRMS (ESI) calculated for C_31_H_51_N_10_O_8_ [M + H]^+^ 691.3886, found 691.3874 (Δ = −1.7 ppm).

Fullerene–Aib (**23**)

C_60_ (51 mg, 0.071 mmol, 1 eq.), compound **22** (49 mg, 0.071 mmol, 1 eq.) and CBr_4_ (26 mg, 0.083 mmol, 1.1 eq.) were dissolved in 25 mL of toluene/CHCl_3_ (4:1). DBU (23 μL, 0.156 mmol, 2.2 eq.) was added to the mixture under nitrogen atmosphere at 0 °C and the reaction was allowed to warm to rt. After 72 h, the solvent was evaporated at reduced pression and the crude mixture was purified by flash chromatography (toluene to recover unreacted C_60_, then DCM/MeOH 92:8) to obtain compound **23** as a red-brown solid (26 mg, 26%) R*_f_* = 0.30.

IR (neat, cm^−1^) 3278, 3068, 1737, 1652, 1532, 1433, 1365, 1216, 1147;

^1^H-NMR (DMSO-*d*_6_, 400 MHz,) δ 1.33 (s, 12H, C(C*H_3_*)_2_), 1.79 (quintuplet, *J* = 7.3 Hz, 4H, COCH_2_C*H_2_*CH_2_), 2.07–2.17 (m, 8H, NHCH_2_C*H_2_*CH_2_N + COC*H_2_*CH_2_CH_2_), 2.59 (t, *J* = 7.3 Hz, 4H, COCH_2_CH_2_C*H_2_*), 3.39 (q, *J* = 6.3 Hz, 4H, NHC*H_2_*CH_2_CH*_2_*N), 3.55 (s, 6H, CO_2_*Me*), 4.40 (t, *J* = 7.1, 4H, NHCH_2_CH_2_*CH_2_*), 7.87 (s, 2H, Ar C*H* triazole); 8.17 (s, 2H, CON*H*C(CH_3_)_2_), 9.52 (t, *J* = 5.7 Hz, 2H, CON*H*CH_2_);

^13^C{^1^H}-NMR (DMSO-*d*_6_, 100 MHz) δ 24.9, 25.4, 25.6, 29.5, 30.2, 34.8, 37.6, 47.4, 52.2, 55.2, 122.3, 137.2–147.0 (C- C_60_), 148.0, 162.4, 171.9, 175.0;

HRMS (ESI) calculated for C_91_H_49_N_10_O_8_ [M + H] ^+^ 1409.3729, found 1409.3705 (Δ = −1.7 ppm).

### 4.4. Double Functionalization of C_60_

Fullerene–man–Aib (**24**)

Fullerene–Man **14** (20 mg, 0.011 mmol, 1 eq.), compound **22** (8 mg, 0.011 mmol, 1 eq.) and CBr_4_ (4 mg, 0.012 mmol, 1.1 eq.) were dissolved in 4 mL of CHCl_3_. DBU (4 μL, 0.25 mmol, 2.2 eq.) was added to the mixture under nitrogen atmosphere at 0 °C and the reaction was allowed to warm to rt. After 24 h the solvent was evaporated at reduced pression and the crude mixture was purified by flash chromatography (DCM/MeOH 9:1) to obtain compound **24** as a dark orange solid (10 mg, 37%) R*_f_* = 0.33.

IR (neat, cm^−1^) 3269, 3062, 2930, 1738, 1666, 1536, 1432, 1369, 1222, 1021, 1005;

UV-Vis *λ*_max_ (nm): 268, 281, 322, 399, 475, 636;

^1^H-NMR (DMSO-*d*_6_, 400 MHz), mixture of rotamers, δ 1.32 (s, 12H, C(C*H_3_*)_2_), 1.70–1.84 (m, 4H, COCH_2_C*H_2_*CH_2_), 1.86–2.20 (m, 32 H in total: 24H COCH_3_ at 1.92, 2.02, 2.03 and 2.11 ppm, overlapped with 4H NHCH_2_C*H_2_*CH_2_N Aib + 4H COC*H_2_*CH_2_CH_2_), 2.54–2.64 (m, 8H, COCH_2_CH_2_C*H_2_* + 4H NHCH_2_C*H_2_*CH_2_N Man), 2.90–3.32 (m, 8H, NHC*H_2_*CH_2_CH*_2_*N), 3.47–3.59 (m, 6H, CO_2_*Me*), 3.94–5.19 (m, 26H in total: 4H triazole-CH_2_O + 8H NHCH_2_CH_2_C*H*_2_-triazole, 14H sugar), 7.71–7.95 (m, 2H, CON*H*C(CH_3_)_2_), 8.18 (br s, 4H, Ar C*H* triazole), 8.76–9.82 (br m, 4H, CON*H*CH_2_);

^13^C{^1^H}-NMR (DMSO-*d*_6_, 100 MHz) δ 14.6, 20.8, 21.0, 21.2, 24.9, 25.4, 25.6, 29.5, 30.2, 34.8, 37.5, 47.5, 52.2, 55.2, 55.4, 60.2, 60.3, 62.3, 65.8, 68.5, 68.6, 69.1, 96.3, 122.3, 125.0, 137.3–147.6 (C–C_60_), 162.2, 170.1, 170.5, 171.8, 171.9, 175.0;

HRMS (ESI) calculated for C_134_H_107_N_18_O_30_ [M + H]^+^ 2447.7395, found 2447.7454 (Δ = +2.4 ppm).

## Data Availability

Not applicable.

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
