# Peer review of "Controlled Decoration of [60]Fullerene with Polymannan Analogues and Amino Acid Derivatives through Malondiamide-Based Linkers"

_molecules, 2022, doi:10.3390/molecules27092776_

Round 1
Reviewer 1 Report
The manuscript molecules-1688607 "Controlled Decoration of [60]Fullerene with Polymannan Analogues and Amino Acid Derivatives through Malondiamide-Based Linkers" by Tanzi et al. describes the development of approach to functionalization of Buckminster fullerene with both polymannan analogues and natural/non-proteinogenic amino acid derivatives. The synthesis was confirmed by 1H, 13C NMR spectroscopy and HRMS. The authors have interesting synthetic results, so I believe that this paper will be of interest to the readers of Molecules.
Questions and comments:
1) Images of all spectra (NMR, HRMS) should be added to the supplementary materials.
2) About synthesis of the compound 24. How did the authors prove which double bond in fullerene 14 reacts with reagent 22? What methods were used to characterize the product 24 structure, except for 1D NMR and HRMS?
3) I recommend comparing the synthetic results obtained in this work with those described in the literature.
2) Minor comments:
- Please check the parts numbering (for example, 4. Materials and Methods should be 3. Materials and Methods).
- Please correct parts Author Contributions, Funding, Acknowledgments, Conflicts of Interest.
- lines 278, 402 "Tetra" should be "tetra".
- "dmso-d6" should be "DMSO-d6"
Author Response
Dear Editor,
- We have revised our article following the referees’ comments.
- Regarding minor comments, edit and typos, the manuscript has been corrected following all the referees’ suggestions
- Regarding the concerns raised by the reviewers, we have made the following changes:
Reviewer 1
- “Images of all spectra (NMR, HRMS) should be added to the supplementary materials”.
Images of 1H, 13C, and HRMS spectra has been included in supplementary materials.
- “About synthesis of the compound 24. How did the authors prove which double bond in fullerene 14 reacts with reagent 22? What methods were used to characterize the product 24 structure, except for 1D NMR and HRMS?”.
In the original paper by Bingel is reported that the cyclopropanation reaction afforded exclusive addition on [6,6] double bonds of the fullerene skeleton (Chem. Ber. 1993, 126, 1957–1959). Moreover, one of advantages of the Hirsch version of cyclopropanation reaction (exploited in our work) is the exclusive formation of [6,6]-bridged adducts (J. Chem. Soc., Perkin Trans. 1, 1997, 1595–1596 and references therein cited). On the basis of these established data and our previous publications (e.g. Chem. Eur. J. 2017, 23, 9462 – 9466) we did not perform further experiments on our compounds.
- “I recommend comparing the synthetic results obtained in this work with those described in the literature”
The introduction section has been properly modified, replacing lines 87–90 of the original manuscript with a short summary of previously reported works involving functionalization of fullerene with diamide-based linkers. Three new bibliographic references have been added. Similarly, the conclusion section was slightly modified to better highlight the obtained synthetic results.
Reviewer 2 Report
This work by Tanzi and colleagues describes the functionalization of fullerene C60 with polymannan and aminoacid derivatives in a tailorable fashion.
1. I think that the work is worthy publishable without further modification in the main text of the manuscript but I recommend major revision because a supplementary material should be provided, including all the NMR spectra. Furthermore, IR spectra of the products should be performed, including their data in the 'Materials and methods section'.
2. There are some typos:
Line 338: a yellow underline should be removed.
Line 448: the authors should write 13C (number as superscript) instead of 13C.
3. The authors should be consistent with the nomenclature. Sometimes they use "13C NMR" or "1H NMR" and also "13C-NMR" and "1H-NMR". They should select only one way to do it.
4. Lines 653-675: the authors should complete these sections properly.
5. For convinience, I suggest to place the "Conclusions" section before "Materials and methods". This will make the paper easir to read.
Author Response
Dear Editor,
- We have revised our article following the referees’ comments.
- Regarding minor comments, edit and typos, the manuscript has been corrected following all the referees’ suggestions
- Regarding the concerns raised by the reviewers, we have made the following changes:
Reviewer 2
- “… a supplementary material should be provided, including all the NMR spectra. Furthermore, IR spectra of the products should be performed, including their data in the 'Materials and methods section'”.
- Done
- “The authors should be consistent with the nomenclature. Sometimes they use "13C NMR" or "1H NMR" and also "13C-NMR" and "1H-NMR". They should select only one way to do it”.
- Done, we used 1H-NMR
- “For convenience, I suggest to place the "Conclusions" section before "Materials and methods". This will make the paper easier to read”.
- We are pleased to receive and follow this suggestion. The conclusions were first placed after “Materials and methods” to follow the structure of the template. Nevertheless, we were not convinced of that choice.
Round 2
Reviewer 1 Report
I thank the authors for answering my questions and improving the manuscript.
This manuscript is a superb and beautiful work.
Author Response
We thank Reviewer 1 for his work
Reviewer 2 Report
I consider the current version of the manuscript suitable for publication.
Author Response
We thank Reviewer 2 for his work